# Long-Term Stability Prediction for Developability Assessment of Biopharmaceutics Using Advanced Kinetic Modeling

**DOI:** 10.3390/pharmaceutics14020375

**Published:** 2022-02-08

**Authors:** Andreas Evers, Didier Clénet, Stefania Pfeiffer-Marek

**Affiliations:** 1Global Research & Development, Discovery Technologies, Merck Healthcare KGaA, 64293 Darmstadt, Germany; 2Bioprocess R&D Department, Sanofi Pasteur, 69280 Marcy l’Etoile, France; 3Global CMC Development, Sanofi-Aventis Deutschland GmbH, Industrial Park Hoechst, 65926 Frankfurt, Germany

**Keywords:** shelf life prediction, advanced kinetic analysis, chemical stability, physico–chemical properties, developability, formulation, in silico modeling, peptides, biologics

## Abstract

A crucial aspect of pharmaceutical development is the demonstration of long-term stability of the drug product. Biopharmaceuticals, such as proteins or peptides in liquid formulation, are typically administered via parental routes and should be stable over the shelf life, which generally includes a storing period (e.g., two years at 5 °C) and optionally an in-use period (e.g., 28 days at 30 °C). Herein, we present a case study where chemical degradation of SAR441255, a therapeutic peptide, in different formulations in combination with primary packaging materials was analyzed under accelerated conditions to derive long-term stability predictions for the recommended storing conditions (two years at 5 °C plus 28 days at 30 °C) using advanced kinetic modeling. These predictions served as a crucial decision parameter for the entry into clinical development. Comparison with analytical data measured under long-term conditions during the subsequent development phase demonstrated a high prediction accuracy. These predictions provided stability insights within weeks that would otherwise take years using measurements under long-term stability conditions only. To our knowledge, such in silico studies on stability predictions of a therapeutic peptide using accelerated chemical degradation data and advanced kinetic modeling with comparisons to subsequently measured real-life long-term stability data have not been described in literature before.

## 1. Introduction

Biopharmaceuticals, such as peptide or antibody drugs, are often provided to patients as an aqueous solution that is applied intravenously, or subcutaneously using an injection device. In addition to its biological efficacy, long-term stability is one of the most crucial parameters that determine the developability of a biopharmaceutical into a commercial drug product since frequent re-supply of drugs causes costs for industry and patients. For registration, the manufacturer must provide information on the stability of the drug product over the shelf life, which is the time from the date of manufacture to the expected viability within its approved product specification while stored under the recommended storage conditions. Instability of a drug product can result in undesired change in performance, adverse side effects and even cause product failures [1,2].

Due to this critical aspect, physical and chemical stability have been recognized as crucial screening and optimization parameters already in early discovery projects by the pharmaceutical industry [3,4,5,6]. Due to their inherent conformational flexibility, therapeutic peptides are generally susceptible to chemical degradation in solution, for example, hydrolysis, oxidation, isomerization or deamidation. In addition, covalent crosslinks between peptides or proteins might result in the formation of covalent dimers and high molecular weight protein (HMWP). Finally, peptides may have a strong amphiphilic character or contain hydrophobic hot spots that trigger self-association and aggregation. Based on its sequence, a peptide has specific vulnerabilities for degradation processes. Both chemical and physical degradation can be significantly affected by parameters of the liquid formulation such as pH, buffering agent, peptide concentration, salt type, ionic strength, or excipients such as co-solutes, preservatives, and surfactants [4,7,8,9,10,11,12,13]. Furthermore, chemical stability is generally affected by external factors, such as temperature, mechanical stress, or interactions with primary packaging materials such as glass and rubbers [14].

Significant efforts have been put into developing and routinely applying predictive in vitro and in silico methods for early compound optimization and developability assessment [15,16,17,18,19]. As part of Sanofi’s integrated strategy for multi-parameter optimization and developability assessment of peptides intended for subcutaneous application via multiple-dose pen devices, we implemented a workflow that includes early analytical profiling considering specific requirements of the target drug product profile. This profiling strategy includes early physical and chemical stability testing under accelerated conditions in relevant tool formulations already in the early discovery phase to design drug candidates with optimal stability properties, as outlined in detail in Refs. [5,20]. As part of this developability assessment, the best candidate molecules are evaluated in an extended physico–chemical property profiling that includes analysis of pH-dependent degradation in buffered solutions and a robotic formulation screening focused on physical stability [21]. Formulations with low aggregation risk are then subjected to chemical stability testing under accelerated conditions (40 °C for 4 weeks). Finally, the most stable formulations are evaluated in an in-depth 3-month stability study for maximum de-risking prior to entry into development. This evaluation includes the variation of different parameters, such as composition of excipients, dose strength, primary packaging materials as well as different temperatures and testing intervals. Experimental analysis comprises different analytical methods, such as high-performance liquid chromatography (HPLC), size-exclusion chromatography (SEC), nephelometry, visual inspection, viscosity measurements, dynamic light scattering (DLS), micro-flow imaging (MFI), Fourier-transform infrared spectroscopy (FTIR), and Thioflavin T (ThT) binding. Finally, we perform long-term chemical stability predictions (for shelf life estimation) based on quantitative attributes such as purity loss and HMWP formation, that are derived from the 3-month accelerated stability study, in order to provide an even more educated risk assessment for the entry into development.

Several approaches have been described that predict the shelf life of liquid dosage forms of biopharmaceutics based on accelerated stability studies [22,23,24,25,26,27,28,29,30]. These are generally performed at elevated temperatures (e.g., 25 and 40 °C), whereas the recommended temperature for long-term storage is usually between 2 and 8 °C for injectable biopharmaceutics. Approaches that are described in the International Conference on Harmonisation (ICH) guidelines [31,32] are generally based on linear or nonlinear regression and statistical modeling through poolability tests [33]. On the other hand, the use of advanced kinetics that considers linear, accelerated, decelerated and S-shaped kinetic profiles might provide an unbiased approach to predict the degradation rates of biopharmaceutics derived from accelerated stability data [34]. Such advanced kinetic descriptions of the experimental degradation data do not require assumptions on the type of kinetics (such as the Arrhenius equation or first-order reactions). Instead, various one-step or two-step models are screened and scored [34]. The derived models showed accurate predictions of long-term stability of complex biopharmaceutics such as vaccines [35,36,37,38,39]. Furthermore, these kinetic models can also be used to predict the effect of temperature excursions and the quality of products in real time during their shipments [40].

In the present study, we applied kinetic-based modeling to estimate the long-term chemical stability of SAR441255, a peptidic unimolecular glucagon-like peptide-1 (GLP-1), glucose-dependent insulinotropic polypeptide (GIP), and glucagon (GCG) receptor triagonist for the treatment of diabetes and obesity [41]. These studies were performed as part of a developability (risk) assessment prior to entry into clinical development. Liquid formulations filled in different containers (primary packaging materials) were tested in accelerated stability studies at temperatures of 5 °C, 25 °C, 30 °C, 37 °C, and 40 °C over a period of up to three months. These data were used to generate a kinetic model for each combination of formulation and primary packaging material to predict the long-term stability under relevant conditions for approval, i.e., two years at 2–8 °C storage plus an additional temperature excursion of 30 °C for 4 weeks (in-use time for multiple-dose drug-device combination). These predictions indicated that the acceptance criteria (peptide content 90–110% of label claim and ≤2% HMWP) would be fulfilled at the end of the targeted shelf life and in-use time, supporting the entry of the drug candidate into the development phase with a low risk for stability issues [41]. Continuation of experimental testing of chemical degradation at the recommended storage temperature of 2–8 °C during the subsequent development phase demonstrated (i) the correctness of the predictions and (ii) the benefit of applying advanced kinetic modeling as part of a developability assessment.

To our knowledge, such in silico studies on stability predictions of a therapeutic peptide using advanced kinetic modeling have not been described in literature before. 

## 2. Materials and Methods

### 2.1. SAR441255 Sequence, Structure, and Background Information

The structure of SAR441255 shown in Figure 1 is based on the selective GLP-1R agonist exendin-4. Following analysis of 3D structural models based on the known X-ray structures of GLP-1, GIP, and GCG in their respective receptors [20,42,43], amino acids of the natural hormones have been introduced into specific positions of exendin-4 to achieve a balanced in vitro activity profile at the different receptors. Further mutations and chemical modifications were introduced in a process of multi-parameter peptide optimization to improve pharmacokinetic properties. In addition, the peptide was modified towards robust physico–chemical properties in an acidic tool formulation at pH 4.5, as described in detail in Ref. [20]. This would allow for daily subcutaneous (s.c.) application of the peptide drug in a multiple-dose pen device, either in a standalone formulation or as co-formulation with Sanofi’s insulin glargine similar to the co-formulation of insulin glargine with the GLP-1R agonist lixisenatide at pH 4.5 in Soliqua^®^ [44].

Chemical degradation of SAR441255 was evaluated in a tool formulation (pH 4.5) and in aqueous buffers covering a pH range from 4.0 to 7.4. Mass spectrometry (MS) analysis showed that the major degradation pathways are (i) deamidation and/or isomerization, (ii) oxidation, and (iii) hydrolysis (see Appendix A). Additionally, covalent dimers (HMWP) were identified among the degradation products. All degradation pathways are clearly pH-dependent. As part of our developability assessment, SAR441255 was then subjected to an extended physico–chemical profiling package including a robotic formulation screening and chemical stability testing under accelerated conditions, resulting in formulations F1–F3 (Table 1). 

In the meantime, SAR441255 has been clinically investigated in healthy subjects, where it demonstrated improved glycemic control during a mixed-meal tolerance test and impacted biomarkers for GCG and GIP receptor activation [41].

### 2.2. Prototype Formulations and Primary Packaging Materials

Solutions for subcutaneous injection were investigated for their stability for a period of up to three months prior to entry into development. Three compositions were selected based on (i) preliminary chemical stability data as well as (ii) an automated formulation screening with focus on aggregation and physical stability, as exemplified in Refs. [20,21,45]. Peptide concentrations of 0.1, 0.5, and 2.5 mg/mL appeared suitable for the intended dosing scheme of phase 1 clinical trials. The lower concentrations were important for the initial dose ramping only. The maximum concentration of 2.5 mg/mL was most relevant for the final drug product consisting of a ready-to-use solution for daily s.c. application using a multiple-dose pen device. It is cost-effective and convenient to provide a significant number of single doses out of one pen-device (weekly or monthly pen) with low injection volumes for minimal injection pain. Thus, prediction of shelf life is desirable for the high concentration of 2.5 mg/mL only. The lower concentrations are not further discussed here.

The investigated compositions are summarized in Table 1. Acetate buffer was used to stabilize the pH value at 4.5 to allow for a potential co-formulation with insulin glargine [44]. Two very different tonicity adjusting agents were explored: (i) glycerol as a non-ionic polyol that does not increase ionic strength above the level imposed by the acetate buffer (formulation F1) and (ii) NaCl as the most common ionic tonicity agent which significantly increases the ionic strength on top of the acetate buffer (formulation F2). Differences in ionic strengths may significantly affect chemical degradation pathways and influence physical stability properties [21]. A further formulation (F3) contains low levels of polysorbate 20 (PS20) as a potential stabilizer for long-term physical stability that may be needed to reduce aggregation risk under mechanical stress (e.g., aggregation at liquid–air or liquid–glass interfaces). All further ingredients were identical in these formulations. L-methionine functions as an antioxidant suppressing oxidation of the peptide at acidic pH as well as HMWP formation (see Appendix A), which is most likely due to oxidative cross-linking of peptide monomers, as recently described in Ref. [46]. The addition of an efficient preservative such as metacresol is mandatory for the administration via multiple-dose devices to safely avoid the growth of microbes that may enter the device from the patient’s skin after application of the first dose [47] during the in-use period. Consideration of more than one prototype formulation was part of our preclinical de-risking strategy to provide at least one stable formulation that can be used without further adjustment for phase 1 clinical trials.

Freshly prepared bulk drug product solutions were filled into different sterilized primary packaging materials (PM1 and PM2, see Table 2) that were relevant for planned clinical trials: (i) 2R iso vials closed with FluroTec^®^ injection stoppers and sealed with flip-off caps (PM1) from which the solution can be removed using a syringe and needle during phase 1 clinical trials, and (ii) siliconized 1.5 mL glass cartridges that are closed with a movable siliconized plunger at one end and a cap with a sealing disk at the other (PM2) and are used in multiple-dose devices for self-injection for late-stage development (phase 2 and 3) and launching the final product. The silicone oil is required to minimize friction between the glass cylinder and the plunger during injection. The FluroTec^®^ stoppers for the 2R iso vials were washed prior to sterilization, removing most of the silicone oil that is used as mold release agent. Since 2R iso vials were always stored in an upright position, the liquid formulation had no contact with potentially spurious silicone oil. Thus, PM1 represents a silicone-free primary packaging with headspace, and PM2 a siliconized primary packaging without headspace. On the one hand, headspace may increase the risk for aggregation caused by the liquid–air interface (agitation stress). In addition, the oxygen within the headspace can influence chemical degradation (e.g., oxidation). On the other hand, silicone oil is known to increase the risk for aggregation or formation of particles within the liquid formulation upon storage. All containers were stored tightly closed. Further details on the primary packaging materials are provided in the Appendix A.

### 2.3. Stability Monitoring

According to the very recently published “good modeling practices” for the generation of kinetic models, it is recommended to produce at least 20–30 experimental stability data points in duplicates, covering (at least) three different incubation temperatures [39]. For the present study, the accelerated stability data of the drug products required for the kinetic analysis were determined after storing the containers in an upright position at 5 ± 3 °C, 25 ± 3 °C, 30 ± 3 °C, 37 ± 3 °C and 40 ± 3 °C in temperature-controlled incubators for up to three months. The time intervals of accelerated degradation measurements were taken as follows: 0, 13, 33, 46, 60, and 90 days at all five temperatures. Due to constraints of API availability prior to the entry into development, we decided at the initiation of the 3-month stability study to run most analytical methods with single-point measurements (no replicates) in favor of a comprehensive coverage of the temperature range, and three formulations (F1–F3), three peptide concentrations, and two primary packaging materials (PM1 and PM2). This described stability study was an internal pilot for maximum risk assessment regarding long-term (physical and chemical) stability with the intension to derive valuable conclusions for future projects. The experimental data for chemical degradation of SAR441255 (minimum of 20–30 points for each drug product) were fitted using the AKTS software [48] as described below. Long-term experimental data were measured after storage of the peptide formulations at the recommended storage temperature of 5 °C for 6 and for 24 months using the remaining backup samples. Additional time points of analysis were requested for both packaging materials PM1 and PM2 (e.g., after 9 and 12 months of storage). However, data could only partially be collected and were not uniform for all drug products due to capacity limitations and changed priorities of the analytical lab. Samples that were originally intended to be analyzed after 12 months were instead used after 24 months of storage at 5 °C (the targeted shelf life period) and subsequent incubation at 30 °C for 28 days (in-use time). Chemical degradation was measured in terms of purity loss (surrogate for the decrease of peptide content) and HMWP formation upon storage (on top of HMWP within drug substance powder) applying HPLC and SEC (see detailed descriptions in the Appendix A).

### 2.4. Advanced Kinetic Modeling and Stability Predictions

The AKTS-Thermokinetics software (version 5.3, AKTS AG, Advanced Kinetics and Technology Solutions, Siders, Switzerland), combining advanced kinetics and statistical analyses of the accelerated stability datasets obtained from 5 °C to 40 °C over a period of three months, was used to identify kinetic models which best describe the rates of purity loss and HMWP increase for each peptide formulation. Except for the use of replicates, our procedure used “good modeling practices” recently recommended for accelerated stability predictions of bioproducts [39,49,50]. Briefly, the software screened and compared the variety of kinetic models, including one-step and two-step models (Equation (1)), describing the reaction progresses (i.e., purity loss and HMWP formation) independent of the complexity of their molecular mechanisms as a function of time and temperature in a manner as previously described [34,35,51].
(1)$$\frac{d\alpha }{dt}=A\times exp\left(-\frac{Ea}{RT}\right){\left(1-\alpha \right)}^{n}\alpha^{m}$$
with α: the reaction progress, *A*: pre-exponential factor, *Eα*: activation energy, *n*: reaction order, *m*: a parameter introduced considering the possible autocatalytic behavior of the reaction.

Long-term predictions of purity loss and HMWP formation as a function of time and temperature were subsequently performed using the best identified models with the optimized kinetic parameters. Subsequently, 99.9% percentile prediction intervals (PIs) were calculated using bootstrap analysis (resampled > 100 times with replacement). Finally, the predicted long-term stability data (purity loss and HMWP formation) after 24 months at 5 °C plus 28 days at 30 °C were experimentally analyzed. To assess prediction accuracy, the difference (as percentage point) between experimentally and predicted purity along with HMWP values were investigated.

## 3. Results

All experimental stability data, including peptide purity and HMWP formation, are provided as Appendix A. The following section presents the kinetic analysis, the long-term predictions with respect to (1) loss of peptide purity, and subsequently, (2) HMWP formation as well as the comparison of these predictions with the experimental long-term stability data. In all cases, the peptide was tested for long-term stability in the six combinations of three formulation and two primary packaging materials shown in Table 3.

### 3.1. Long-Term Prediction of Purity Loss

For each combination of the three peptide formulations and two primary packaging materials, a large variety of models from the simplest (zero and first-order) to the more complex were screened to fit the three-month experimental data using AKTS. The software compared and ranked these models. The best models were selected based on their Akaike and Bayesian information criterion (AIC and BIC) scores, and Residual Sum of Squares (RSS) values. Equations (2)–(7), Figure 2 and Table 4 show the best models obtained for all six cases.
(2)$$\frac{d\alpha }{dt}=exp\left(16.1\right)\times exp\left(-\frac{85.8E3}{RT}\right)\times {\left(1-\alpha \right)}^{4}\times \alpha^{0.2}$$
(3)$$\frac{d\alpha }{dt}=exp\left(15.7\right)\times exp\left(-\frac{86.6E3}{RT}\right)$$
(4)$$\frac{d\alpha }{dt}=exp\left(16.7\right)\times exp\left(-\frac{89.3E3}{RT}\right)\times \left(1-\alpha \right)$$
(5)$$\frac{d\alpha }{dt}=exp\left(17.7\right)\times exp\left(-\frac{90.0E3}{RT}\right)\times {\left(1-\alpha \right)}^{4}\times \alpha^{0.2}$$
(6)$$\frac{d\alpha }{dt}=exp\left(16.7\right)\times exp\left(-\frac{89.5E3}{RT}\right)$$
(7)$$\frac{d\alpha }{dt}=exp\left(16.1\right)\times exp\left(-\frac{85.1E3}{RT}\right)\times {\left(1-\alpha \right)}^{6}\times \alpha^{0.3}$$

As shown in Table 4 and Figure 2, the predicted stability data for all 6 combinations of formulation–packaging materials (Table 3) were in full agreement with our internal acceptance criteria (>90% peptide purity at the end of shelf life and in-use time) and supported the decision for entry into clinical development. The experimental long-term stability data that were later experimentally determined at the end of shelf life (24 months at 5 °C) and in-use time (28 days at 30 °C after 24 months stored at 5 °C) confirmed that the data had been correctly predicted within the 99.9% prediction interval (PI); see Figure 2. In all six cases, one-step models were able to fit the experimental data. Only for two out of the six formulation–packaging combinations, zero-order kinetics were observed (Equations (3) and (6)), whereas the degradation for the other three cases is described by higher reaction orders. Indeed, such higher reaction orders have often been observed for other biologicals, such as antibodies and vaccines under specific conditions, e.g., Refs. [34,35,36,37,51,52].

### 3.2. Long-Term Prediction of HMWP Formation

As outlined above, numerous models were screened to fit the three-month experimental data, and the best models were selected based on their AIC and BIC scores, and RSS values. Applied kinetic models, curve progressions and fit of experimental data points (including long-term data) by simulated curves are presented in Equations (8)–(13), Table 5, and Figure 3.
(8)$$\frac{d\alpha }{dt}=0.67\times exp\left(17.8\right)\times exp\left(-\frac{87.5E3}{RT}\right)+0.33\times exp\left(4.0\right)\times exp\left(-\frac{48.2E3}{RT}\right)\times {\left(1-\alpha \right)}^{0.5}$$
(9)$$\frac{d\alpha }{dt}=exp\left(13.0\right)\times exp\left(-\frac{74.6E3}{RT}\right)\times {\left(1-\alpha \right)}^{}$$
(10)$$\frac{d\alpha }{dt}=exp\left(10.1\right)\times exp\left(-\frac{66.5E3}{RT}\right)\times {\left(1-\alpha \right)}^{}$$
(11)$$\frac{d\alpha }{dt}=0.85\times exp\left(19.4\right)\times exp\left(-\frac{95.0E3}{RT}\right)\times {\left(1-\alpha \right)}^{}+0.15\times exp\left(4.4\right)\times exp\left(-\frac{46.4E3\;}{RT}\right)\times {\left(1-\alpha \right)}^{3}$$
(12)$$\frac{d\alpha }{dt}=0.04\times exp\left(4.3\right)\times exp\left(-\frac{46.7E3}{RT}\right)\times {\left(1-\alpha \right)}^{2}+0.96\times exp\left(16.1\right)\times exp\left(-\frac{90.2E3}{RT}\right)\times {\left(1-\alpha \right)}^{}$$
(13)$$\frac{d\alpha }{dt}=0.97\times exp\left(14.3\right)\times exp\left(-\frac{85.7E3}{RT}\right)\times {\left(1-\alpha \right)}^{4}+0.03\times exp\left(6.7\right)\times exp\left(-\frac{51.2E3}{RT}\right)\times {\left(1-\alpha \right)}^{2}$$

In all cases, the predictions suggested that the acceptance criterion at the end of the targeted shelf life (two years storage at 5 °C) and in-use time (28 days at 30 °C) would be fulfilled (<2% HMWP formation) and supported entry into clinical development. As observed by experimental determination of the real-time stability after storage for two years (at 5 °C), the long-term stability data had even been correctly predicted within the 99.9% PI in all six cases (see Figure 3a). Inspection of the reaction plots (Figure 3) and corresponding rate equations (Equations (8)–(13)) clearly demonstrates that HMWP formation does not linearly increase over time. Furthermore, in four out of six cases, HMWP formation is described by two-step kinetic models (Equations (8), (11)–(13)), which is in agreement with previous studies for peptides or antibodies [53,54]. Due to this obvious non-linearity of the kinetic profile, it was essential to gather multiple experimental data points from elevated temperatures for deriving the rate equations that realistically describe the complex process of HMWP formation to predict stability under long-term conditions (e.g., two years at 5 °C), especially at higher temperatures (e.g., for the assessment of the in-use time at 30 °C).

For the prediction of HMWP formation after long-time storage for two years at 5 °C plus an additional temperature excursion of 28 days at 30 °C (in-use time), the experimental value was correctly predicted in four out of six cases within the 99.9% PI (see Figure 3b). 

## 4. Discussion

Shelf life and long-term stability of biopharmaceutical drug products are crucial aspects of pharmaceutical development. Advanced kinetic modeling based on accelerated stability data is a constantly evolving approach for the prediction of long-term stability. Although such modeling approaches are not yet required for New Drug Submission according to official guidelines, many regulatory agencies are now appreciative of their value in dossier submissions.

In this genuinely prospective and predictive study that included the measurement of long-term stability data, we demonstrated the scope and usefulness of kinetic modeling for long-term stability predictions on SAR441255. This study describes, to our knowledge, the first application of advanced kinetic modeling on a therapeutic peptide, where the effects of different formulations and packaging materials on long-term stability were evaluated head-to-head and subsequently verified by real-life long-term experimental data. The predictions, derived from 3-month accelerated stability data, suggest that all three formulations of SAR441255 investigated in two different packaging materials are sufficiently stable over the targeted shelf life, including a storing period of two years at 5 °C, and the terminal in-use period of 28 days at 30 °C. These predictions, made in the preclinical phase as part of a developability assessment, served as a crucial decision parameter for fast project progression into clinical development. As experimentally verified after two years storage under long-term conditions, all predictions were mainly within the 99.9% PI with a maximum deviation of less than only 1%. Since prototype formulations containing glycerol as tonicity agent showed a lower risk for aggregation than those with NaCl (data not shown), Combi_F1-PM1 was chosen for clinical phase 1 studies using vial and syringe for s.c. injections and Combi_F1-PM2 for clinical trials using a multiple-dose pen device. Consequently, these studies offered stability insights within three months that would have taken two years (plus 28 days for in-use stability) to determine under long-term storage conditions and thereby provided a significant acceleration and de-risking of the project already before entry into clinical development. 

Very recently, Kuzman et al. [25] successfully used Arrhenius-based kinetics to predict long-term stability of IgG1, IgG2, and fusion proteins. For some key stability attributes such as sum of aggregates by SEC, accurate predictions of long-term stability were obtained using first-order kinetics. However, other attributes such as the amount of basic forms as determined by cation exchange chromatography (CEX) could not to be fitted by first-order kinetics, requiring more sophisticated kinetic models to best describe experimental data at 5 °C, 25 °C, and 40 °C. In contrast to the use of the Arrhenius equation for reaction mechanisms, advanced kinetic models do not require assumptions on the type of kinetics. Instead, a phenomenological mathematical model is derived for fitting the reaction progress, without any mechanistic basis, allowing to model processes that show Arrhenius and non-Arrhenius behavior.

For SAR441255, all predicted and experimentally observed data of purity loss after long-term storage were highly similar, indicating that the different formulations and packaging materials did not significantly influence the degradation process. The degradation plots (Figure 2a) indicate a rather linear purity loss over time at low temperatures with Arrhenius-like behavior. In contrast, HMWP formation (Figure 3a) did not linearly increase over time and shows a more complex reaction progress according to Equations (8)–(13). HMWP formation of peptides generally originates from covalent crosslinking of reactive or degraded (e.g., oxidized) species of two peptide monomers [46] and therefore depends on the concentration and reactivity of both partners. Despite this mechanistic complexity, the long-term stability could be accurately predicted with the kinetic models derived from the accelerated stability studies. Notably, within the project a preliminary linear extrapolation of the first HMWP data points was done after the initial phase of accelerated stability testing (considering only data points at 5 °C after 4 weeks) as a first estimate for the degree of HMWP formation after two years (at 5 °C). As obvious from visual inspection of these data points in Figure 3a, this linear extrapolation raised the concern that the rate of HMWP formation would be significantly outside the acceptance criteria (>2%) after long-term storage. However, as (1) suggested by long-term stability predictions based on advanced kinetic models after finalization of the accelerated stability studies and (2) finally verified by real-time measurements, the degree of HMWP formation was significantly below 2% at the end of shelf life at 5 °C and even after the subsequent in-use time (28 days at 30 °C). Detailed comparison of the experimental long-term data reveals a slightly higher HMWP formation in identical formulations for PM2 compared with PM1, suggesting a potential influence of the siliconized surfaces and/or missing headspace on HMWP formation. 

Such non-linear degradation over time, as detected for HMWP formation of SAR441255, is also often observed for several (e.g., aggregation related) quality attributes of vaccines and therapeutic antibodies [25,34,35,36,37,49,51,52,53,54], underlining the benefit of screening various kinetic models for biopharmaceutics. With appropriate kinetic models that accurately describe the degradation progress as a function of time and temperature, various applications emerge, such as estimation of shelf life [34,35,36,37,38], accurate degradation-based predictions for products during various types of temperature excursions [35,38,49], ranking of formulations [37,39,49], batch-to-batch comparisons and real-time shelf-life monitoring of products during shipments [39,40,49]. 

The findings of this study should, however, be considered in the light of some limitations and recommendations for future studies. (1) Due to limited API availability, we did not measure stability data points in duplicate or triplicate, which might explain that the predicted HMWP formation after long-time storage for two years at 5 °C plus an additional temperature excursion of 28 days at 30 °C (in-use time) was slightly outside the 99.9% PI in two cases (see Figure 3b). Therefore, we suggest, in agreement with previous recommendations [39], to perform future analytical measurements at least in duplicate, which might aid experimental outlier detection and further increase prediction accuracy. (2) As mentioned in the introduction, chemical stability of a peptide depends on its sequence and parameters such as the pH of the liquid formulation, further formulation ingredients, peptide concentration and possible interactions with the primary packaging material. Consequently, kinetic models for long-term stability predictions are generally only applicable to the set of parameters used during the accelerated stability studies, i.e., the obtained models will generally not be applicable to other pH values or peptide concentrations. Therefore, it is highly important to ensure that the parameters chosen for the accelerated stability studies are most relevant for the use in clinical trials and for the intended commercial drug product. (3) The present study is the first report of long-term stability predictions on a therapeutic peptide in liquid formulations. Of course, further studies will be required to demonstrate general applicability of this approach for therapeutic peptides. However, successful applications of long-term predictions using advanced kinetic modeling for vaccines [34,35,36,37,38,39,40] and antibodies using Arrhenius kinetics [25] suggest that these approaches are generally applicable for a broader range of modalities. The AKTS software can, in principle, operate on any kind of data that follow a reaction progress without the need to know the exact molecular mechanism and kinetic model. (4) Finally, we did not investigate in detail in how far prediction accuracy would be impacted by focusing experimental measurements only on specific temperatures or timepoints during the accelerated stability studies. In the present study, measurements were done at 5 °C, 25 °C, 30 °C, 37 °C, and 40 °C for up to three months. According to the recently published recommendations for “good modeling practices” for the generation of kinetic models [39], it might be more feasible to produce experimental stability data points (as duplicates) that cover only three different incubation temperatures (e.g., 5 °C, 25 °C, and 40 °C).

In conclusion, the present study demonstrates that using prior knowledge gained from accelerated stability studies in combination with advanced kinetic modeling can provide accurate predictions of a drug product’s long-term stability under the recommended storage conditions and thereby accelerate and de-risk pharmaceutical development. As additional benefit, kinetic models allow to predict degradation rates for a drug product being exposed to successive temperature excursions during storage, shipment and distribution and might therefore significantly aid life cycle management of pharmaceutical products. 

## Figures and Tables

**Figure 1 pharmaceutics-14-00375-f001:**
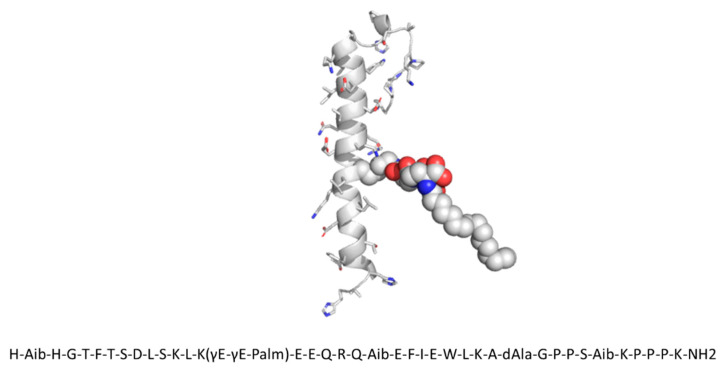
Amino acid sequence and modeled three-dimensional structure of the peptide. Aib is a 2-aminoisobutyric acid residue and dAla is a D-alanine. Residue 14 is modified by addition of a C16 fatty acid (palmitic acid) at the ε-amino group of lysine using two γ-glutamic acid spacers.

**Figure 2 pharmaceutics-14-00375-f002:**
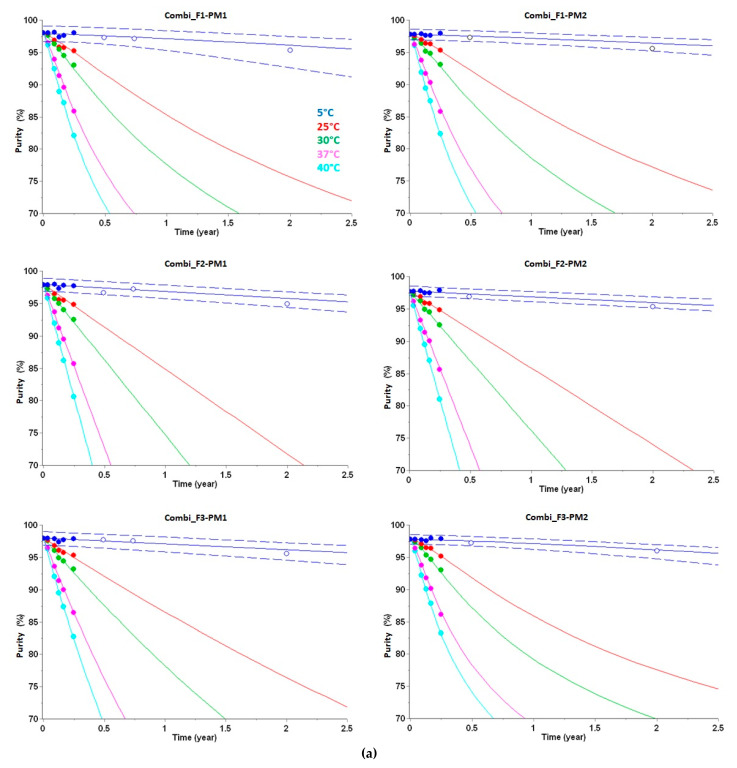
Predictions of peptide purity (in %) for three different formulations in two different packaging materials. (**a**) Purity predictions are displayed as lines for up to 2.5 years at 5 °C (blue), 25 °C (red), 30 °C (green), 37 °C (pink) and 40 °C (cyan). High-Performance Liquid Chromatography (HPLC) data used for kinetic modeling are displayed as filled circles. Experimental long-term stability data points, that were not used for model building, are displayed as open circles. Data after 9 months were only determined for PM1 due to capacity limitations of the analytical lab. At 5 °C, purity predictions are shown with predictive bands representing 99.9% PI (dashed lines). Additional experimental data obtained after two years, not used for kinetic modeling, are displayed as open circles. (**b**) Purity of peptides predicted from best kinetic models (blue solid lines) during temperature excursions (28 days at 30 °C) outside the cold chain (two years). Purity predictions are shown with predictive bands representing 99.9% PI (dashed lines). Experimental data determined at the end of temperature excursions, and not used for kinetic modeling, are displayed as open blue circles.

**Figure 3 pharmaceutics-14-00375-f003:**
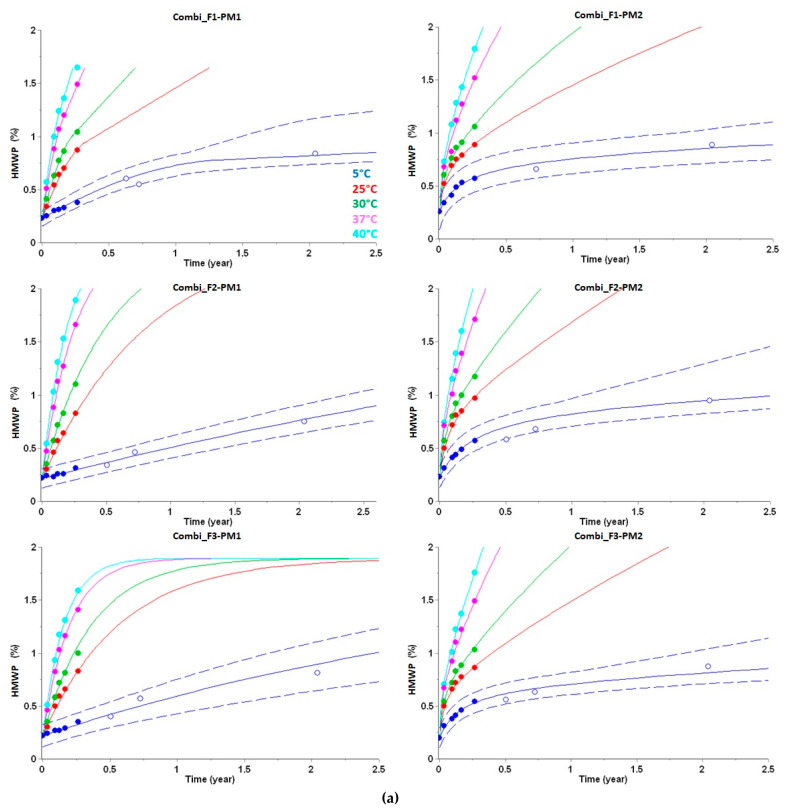
Predictions of high molecular weight protein (HMWP) formation (in %) for three different peptide formulations in two different packaging materials. (**a**) HMWP predictions are displayed as lines for up to 2.5 years at 5 °C (blue), 25 °C (red), 30 °C (green), 37 °C (pink) and 40 °C (cyan). HMWP data used for kinetic modeling are displayed as filled circles. Experimental long-term stability data points, that were not used for model building, are displayed as open circles. These data points beyond 3 months are not uniform for all drug products due to capacity limitations and organizational aspects of the analytical lab. At 5 °C, HMWP predictions are shown with predictive bands representing 99.9% PI (dashed lines). Additional experimental data obtained after two years, and not used for kinetic modeling, are displayed as open circles. (**b**) HMWP of peptides predicted from the best kinetic models during temperature excursions outside the cold chain. HMWP predictions from customized temperature excursions are displayed as lines for 28 days in an incubator set at 30 °C (red, solid line). HMWP predictions are shown with predictive bands representing 99.9% PI (dashed lines). Experimental data determined at the end of temperature excursions, and not used for kinetic modeling, are displayed as open circles.

**Table 1 pharmaceutics-14-00375-t001:** Composition of prototype formulations (F1, F2, F3) for three-month stability study.

Ingredient	F1 (mg/g)	F2 (mg/g)	F3 (mg/g)
acetic acid	0.88	0.88	0.88
sodium acetate trihydrate	0.51	0.51	0.51
peptide	2.5	2.5	2.5
glycerol	20.4		20.4
NaCl		6.47	
L-methionine	3	3	3
metacresol	2.7	2.7	2.7
polysorbate 20			0.01
NaOH	Ad pH 4.5	Ad pH 4.5	Ad pH 4.5
water for injection	Ad 1 g	Ad 1 g	Ad 1 g

**Table 2 pharmaceutics-14-00375-t002:** Primary packaging materials (PM1, PM2) used for storage.

Identifier	Container	Injection Stopper/Cap	Plunger Stopper	Siliconized Surfaces
PM1	2R iso vial, clear glass	13 mm grey bromobutyl rubber with Flurotec^®^ coating + 13 mm flip-off cap	-	none
PM2	1.5 mL cartridge, clear glass, baked in silicon	7.5 mm aluminum flanged cap with laminated sealing disc	6 mm bromo-butyl siliconized rubber	container + plunger

**Table 3 pharmaceutics-14-00375-t003:** Combinations of three different peptide formulations and two different packaging materials that were used for experimental and in silico assessment of accelerated and long-term stability. Details about the formulations (F1, F2, F3) and packaging materials (PM1, PM2) are provided in Table 1 and Table 2.

Name	Formulation	Packaging Material (PM)	Corresponding Kinetic Equations for Long-Term Prediction
Combi_F1-PM1	F1	PM1	(2), (8)
Combi_F2-PM1	F2	PM1	(3), (9)
Combi_F3-PM1	F3	PM1	(4), (10)
Combi_F1-PM2	F1	PM2	(5), (11)
Combi_F2-PM2	F2	PM2	(6), (12)
Combi_F3-PM2	F3	PM2	(7), (13)

**Table 4 pharmaceutics-14-00375-t004:** Experimental vs. predicted values of peptide purity as measure for long-term chemical stability for three different peptide formulations in two different packaging materials combined into six drug products after storage for two years at 5 °C and after a subsequent in-use period of 28 days at 30 °C. The corresponding plots and rate equations are provided in Figure 2 and Equations (2)–(7).

	Purity (%) after Two Years at 5 °C			Purity (%) after Two Years at 5 °C Plus 28 Days at 30 °C		
Drug Product	Experimental	Predicted	Δ	Experimental	Predicted	Δ
Combi_F1-PM1	95.3	96.1	0.8	94.6	94.3	−0.3
Combi_F2-PM1	94.9	95.8	0.9	93.9	94.0	0.1
Combi_F3-PM1	95.6	96.2	0.6	94.7	94.5	−0.2
Combi_F1-PM2	95.6	96.4	0.8	94.4	94.8	0.4
Combi_F2-PM2	95.3	96.0	0.7	93.7	94.3	0.6
Combi_F3-PM2	96.0	96.1	0.1	94.4	94.4	0.0

**Table 5 pharmaceutics-14-00375-t005:** Experimental vs. predicted increase of HMWP (in %) for three different peptide formulations in two different packaging materials combined into six drug products after storage for two years at 5 °C and after a subsequent in-use period of 28 days at 30 °C. The corresponding plots and rate equations are provided in Figure 3 and Equations (8)–(13).

	HMWP Increase (%) after Two Years at 5 °C			HMWP Increase (%) after Two Years at 5 °C Plus 28 Days at 30 °C		
Drug Product	Experimental	Predicted	Δ	Experimental	Predicted	Δ
Combi_F1-PM1	0.84	0.82	−0.02	1.13	0.93	−0.10
Combi_F2-PM1	0.75	0.77	0.02	1.16	1.00	−0.16
Combi_F3-PM1	0.81	0.90	0.09	1.12	1.08	−0.04
Combi_F1-PM2	0.89	0.86	−0.03	1.21	0.97	−0.24
Combi_F2-PM2	0.95	0.95	0.00	1.30	1.07	−0.23
Combi_F3-PM2	0.87	0.81	−0.06	1.12	0.94	−0.18

## Data Availability

The data presented in this study are available upon request from the corresponding author.

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
