# Peer review of "Long-Term Stability Prediction for Developability Assessment of Biopharmaceutics Using Advanced Kinetic Modeling"

_pharmaceutics, 2022, doi:10.3390/pharmaceutics14020375_

Round 1

Reviewer 1 Report

The manuscript describes the results of in vitro study on the stability of new peptide for the treatment of diabetes and obesity. The data obtained have some practical value, however, the novelty of the research performed is rather limited and several issues need clarification:

The study was focused on one peptide concentration. It is not certain whether the predictions made can be extrapolated to other initial concentrations of the peptide.

The single-point measurements and using only one peptide in this study are also rather problematic. It is not clear,  based on the results obtained, whether a similar approach may be used for other peptides or antibodies.

The long-term studies for all temperatures tested should have been performed to validate the models used.

It was not explained why the formulation or the packing material influenced the reaction order so strongly (e.g. it changed from 4 to 0 in some cases) and what are the mechanisms of the fourth- or sixth-order reactions.

The mechanisms of peptide degradation/HMWP formation have not been explained.

 It has not been indicated in the conclusion which peptide formulation and packaging material  is  the most appropriate for the phase I clinical trial.

The effects of other (than temperature) factors such as vigorous shaking or other forms of shear during shipment, the effect of light, or humidity  were not discussed (or tested).

The recommendations for the design of accelerated stability studies (lines 333-357) are not supported by the results.

Table 3 Equations 2g and 3g have not been presented in the manuscript.

Reviewer 2 Report

This manuscript presents a case study where chemical degradation of a therapeutic peptide in different formulations in combination with primary packaging materials was analyzed under accelerated conditions to derive long-term stability predictions using advanced kinetic modeling.

A few concerns for the authors.

  1. The peptide drug used in this case study is an exendin-4 based agonist for the treatment of diabetes and obesity and is formulated in pH 4.5. Will the pH value be considered while predicting the long-term stability? Can the advanced kinetic modeling to be used in other cases when pH value is different?
  2. For figure 2, y axis of each graph needs to be more specific, not just (%).
  3. Same for figure 3, y axis needs to be more specific as it is a little confusing to the readers.
  4. In the discussion session, while the authors tested three different formulations in combination with two different primary package materials, no discussion has been found about these differences regarding to the long-term stability.
  5. In this case study, the authors used five different temperatures to test the long-term stability. However, the only the data under 5oC were used for comparation between the prediction and the real data. More explanations about the other temperatures used need to be provided and the results to be interpreted.
